# Interaction Effect between Physical Activity and the *BDNF* Val66Met Polymorphism on Depression in Women from the PISMA-ep Study

**DOI:** 10.3390/ijerph19042068

**Published:** 2022-02-12

**Authors:** Juan Antonio Zarza-Rebollo, Esther Molina, Elena López-Isac, Ana M. Pérez-Gutiérrez, Blanca Gutiérrez, Jorge A. Cervilla, Margarita Rivera

**Affiliations:** 1Department of Biochemistry and Molecular Biology II, Faculty of Pharmacy, University of Granada, 18071 Granada, Spain; jazarza@ugr.es (J.A.Z.-R.); lopezisac@ugr.es (E.L.-I.); ampergut@correo.ugr.es (A.M.P.-G.); mrivera@ugr.es (M.R.); 2Institute of Neurosciences, Biomedical Research Centre, University of Granada, 18016 Granada, Spain; blancag@ugr.es (B.G.); jcervilla@ugr.es (J.A.C.); 3Department of Nursing, Faculty of Health Sciences, University of Granada, 18071 Granada, Spain; 4Department of Psychiatry, Faculty of Medicine, University of Granada, 18016 Granada, Spain; 5Mental Health Service, University Hospital San Cecilio, 18016 Granada, Spain

**Keywords:** depression, physical activity, *BDNF*, Val66Met, rs6265

## Abstract

The relationship between depression and the Val66Met polymorphism at the brain-derived neurotrophic factor gene (*BDNF*), has been largely studied. It has also been related to physical activity, although the results remain inconclusive. The aim of this study is to investigate the relationship between this polymorphism, depression and physical activity in a thoroughly characterised sample of community-based individuals from the PISMA-ep study. A total of 3123 participants from the PISMA-ep study were genotyped for the *BDNF* Val66Met polymorphism, of which 209 had depression. Our results are in line with previous studies reporting a protective effect of physical activity on depression, specifically in light intensity. Interestingly, we report a gene-environment interaction effect in which Met allele carriers of the *BDNF* Val66Met polymorphism who reported more hours of physical activity showed a decreased prevalence of depression. This effect was observed in the total sample (OR = 0.95, 95%CI = 0.90–0.99, *p* = 0.027) and was strengthened in women (OR = 0.93, 95%CI = 0.87–0.98, *p* = 0.019). These results highlight the potential role of physical activity as a promising therapeutic strategy for preventing and adjuvant treatment of depression and suggest molecular and genetic particularities of depression between sexes.

## 1. Introduction

Depression is a major public health problem, affecting more than 264 million people worldwide, with a higher prevalence in women. It is a leading cause of disease burden and years of disability [1,2], and is also associated with excess mortality [3]. Even though novel pharmacological choices have arisen in the past decades, there still remains a lack of efficiency in pharmacological treatment. In this aspect, there is a rate of non-response to the first election of up to two-thirds of the patients, and between 15–33% in multiple interventions [4]. Therefore, the use of antidepressants has prompted a long-standing debate regarding their effectiveness compared to placebo [5,6]. In an extensive meta-analysis of published and unpublished clinical trials, a small drug-placebo difference directly related to the initial severity of depression was reported [7]. 

Consequently, novel approaches, such as physical activity, have emerged as potential therapeutic strategies, with a promising effect both decreasing the symptomatology of depression [7,8], and in its prevention [9,10,11]. Remarkably, among other variables, it is plausible that sex may have a differential role in the impact of physical activity on depression, according to different studies. In this regard, it has been reported that physical activity leads to more extensive benefits on executive processes in women than in men in a healthy population [12]. These differences remain unclear when a population with depression is considered. Some studies have reported greater differences in the relationship between physical activity and the improvement in depressive symptoms in men [13,14], whereas others have reported similar results in women [15,16,17]. Nonetheless, recent studies with extensive samples and meta-analyses did not find that these differences between sexes were statistically significant [18,19,20]. 

One of the mechanisms suggested to be involved in the relationship between depression and the practice of physical activity is the brain-derived neurotrophic factor (*BDNF*), a neurotrophin related to key brain processes at molecular and functional stages, such as growth and survival of neurons [21], learning and memory [22]. The practice of physical activity has been associated with an increase of BDNF and genetic expression profiles promoting brain plasticity in animal models [23], with similar results in humans (for a review, see [24]). In this respect, current literature points towards a transient increase in peripheral BDNF concentrations after acute aerobic exercise, but not after strength exercise [25], and also an increase in resting concentrations of this neurotrophin after interventions with aerobic—but not resistance—physical activity [26]. Nonetheless, a recent meta-analysis of exercise interventions in depression was not able to extract any conclusive results from the available literature, highlighting the diversity of the studies included in the systematic review [27].

The Val66Met polymorphism, also known as rs6265, is a functional polymorphism within the promoter of the *BDNF* gene. The Met allele of this polymorphism has been associated with impaired regulation of secretion and intracellular trafficking of the BDNF protein in hippocampal neurons [28]. This risk allele has also been associated with modifying the protein function of BDNF in humans [29,30], and although it has been extensively studied in relation to depression and physical activity, the results remain inconclusive. The most recent meta-analyses investigating the association between depression and the Val66Met polymorphism found that this polymorphism does not confer risk for depression [31,32,33]. The effect of physical activity on serum BDNF has also been reported to depend on the Val66Met genotype, being that the difference in serum BDNF is significantly higher after physical activity exclusively in Val/Val participants [34,35]. However, this association could not be replicated in a subsequent study [36]. A more recent systematic review included studies assessing the association of this genotype with the effect of physical activity in diverse cognitive domains, concluding that the available evidence is too limited to draw conclusions [37]. Moreover, the association of this polymorphism as a mediator of the effect of physical activity on depressive symptoms has been assessed in several studies, also showing conflicting results. In a cross-sectional study comparing genetic susceptibility for depression between athletes and non-athletes, the Val/Val genotype was associated with a higher risk of reporting depressive symptoms only in non-athletes [38]. A recent observational study shows that physical activity moderates the association between depression and cognitive function, obtaining better results in different tasks in Met allele carriers [39]. In contrast, another observational study did not find a relationship between the Val66Met genotype, physical activity, and depressive symptoms [16]. Studies that included an intervention with physical activity in patients with depression have reported a greater decrease in depressive symptoms after the intervention, only in men with the Met allele of the Val66Met polymorphism [40], or in Met allele carriers who did not inform exposure to childhood adversity [41]. In adolescent women, a protective effect against depressive symptoms was found with higher levels of physical activity only in Met allele carriers [42]. Given the inconclusive results shown in the literature, we aim to investigate the involvement of the *BDNF* Val66Met polymorphism with depression and physical activity in a large sample of community-dwelling adults.

## 2. Materials and Methods

### 2.1. Study Design

The PISMA-ep is a cross-sectional epidemiological study performed in a representative cohort of community-dwelling adults aiming to analyse the prevalence of psychiatric disorders and their correlates in Andalusia (south of Spain). The three main objectives of the PISMA-ep were: (1) to estimate the prevalence of common mental disorders in Andalusia, (2) to explore the associations existing between social, psychological and genetic factors with mental disorders, and (3) to gather data from an extensive cohort that could be used as the basis for further prospective studies. A more detailed description of the methodology and procedure of this study has been published elsewhere [43]. 

### 2.2. Sample

Randomly selected adults aged 18–75 years old living in all 8 provinces of Andalusia for at least a year were asked to participate in the PSIMA-ep study. We undertook a multistage sampling using different standard stratification levels utilizing a door-knocking approach. We excluded those individuals with an illness that precluded the completion of the interview, not speaking Spanish fluently, suffering from severe cognitive impairment or intellectual disability, and usually residing in an institution. 

### 2.3. Measures

Neuropsychiatric measures: The DSM-IV diagnosis of major depression was ascertained using the Spanish version of the Mini-International Neuropsychiatric Interview (MINI) [44,45]. The MINI is a brief diagnostic structured interview that provides Axis I DSM-IV and ICD-10 compatible diagnoses for 16 mental disorders, including major depression. The MINI has obtained satisfactory psychometric properties, with good rates of validity and reliability on community-based populations [46,47]. This interview was conducted by a team of fully trained psychologists.

Anthropometric measures: For each participant, self-reported height and weight were obtained to calculate their body mass index (BMI) using the formula: weight in kilograms divided by height in meters squared (kg/m^2^). Participants were grouped into four categories, following WHO criteria [48]: underweight (BMI < 18.5 kg/m^2^), normal weight (BMI 18.5–24.99 kg/m^2^), overweight (BMI 25.0–29.99 kg/m^2^) and obesity (BMI ≥ 30 kg/m^2^).

Physical activity: This information was gathered from a questionnaire including 3 questions about whether the participant practised any physical activity, the number of hours per week of physical activity and the intensity of the activity. The intensity was classified based on the Metabolic Equivalents of Task or METs (2 METs = two times the amount of oxygen consumed at rest) as light (<3 METs), moderate (3–5 METs) or vigorous (≥6 METs).

Genotyping analysis: A biological sample was obtained from each participant with an Oragene^®^ saliva DNA (OG-500; DNA Genotek Inc., Kanata, ON, Canada) collection kit. The Oragene^®^ Saliva Collection Kit protocol was used for DNA extraction. The original DNA samples were prepared to be stored at −80 °C in matrix plaque format. DNA quantification was measured using the Infinite^®^ M200 PRO Multimode Microplate Reader (Tecan, Research Triangle Park, NC, USA). Genotyping of the *BDNF* Val66Met polymorphism was assessed using TaqMan^®^ StepOnePlusTM Real-Time PCR System (Applied Biosystems, Foster City, CA, USA) following the manufacturer’s instructions. The system software was used to analyse raw data.

### 2.4. Statistical Analyses

All statistical analyses were performed using R (version 4.0.3) [49]. The R package ‘HardyWeinberg’ was used to test Hardy–Weinberg equilibrium (HWE) and distribution of genotypes, both in the entire sample and in depression cases and controls, using Pearson’s chi-squared tests [50,51]. 

We performed descriptive exploratory analyses to survey how dependent and independent variables were distributed and then explored univariable associations, considering parametric or non-parametric significance tests when required.

Logistic regression models were performed to explore the associations between: (1) physical activity variables (a. whether the participant practised physical activity, b. number of hours of physical activity and c. intensity of the activity) and depression and (2) the Val66Met genotype and depression. A dominant genetic model was assumed, due to the limited sample size. Finally, we assessed the interaction between the genetic (Val66Met genotype) and environmental (the practice of physical activity) variables, using multivariate logistic regression models. We estimated the probabilities for depression by combining the Val66Met genotype (Val/Val homozygous vs. Met allele carrier) and physical activity (binomial, number of hours or intensity). All the association and interaction analyses were performed both crudely and including sex, age and BMI as covariates.

## 3. Results

### 3.1. Description of the Sample

From the 4507 PISMA-ep total sample, 4286 (95.1%) participants accepted to provide a saliva sample for the genetic studies. From those, 3194 (74.52%) were genotyped for the *BDNF* Val66Met polymorphism. A total of 71 (2.22%) participants with BMI under 18.5 were excluded from the analyses. The final sample consisted of 3123 community-based adults, of which 209 were cases with depression (6.69%) (Table 1).

The characteristics of the sample including the frequencies of the independent variables analysed, both genotypic (Val66Met polymorphism) and phenotypic (practice of physical activity, number of hours and intensity) are detailed in Table 1.

### 3.2. The BDNF Val66Met Polymorphism and Depression

There was no significant association between carrying the Met allele of the Val66Met polymorphism and depression, in the total sample in crude analyses nor after adjusting for sex, BMI and age (OR = 1.04, 95%CI = 0.77–1.39, *p* = 0.81). Furthermore, these results were not significant neither in women (OR = 1.05, 95%CI = 0.73–1.50, *p* = 0.78) nor in men (OR = 1.01, 95%CI = 0.59–1.70, *p* = 0.974) after adjusting for age and BMI. The frequencies of the Val66Met genotypes are detailed in Table 2.

### 3.3. Depression and Physical Activity

We found a statistically significant protective effect against depression in those participants reporting any physical activity, which remained significant after adjusting for covariates (age and BMI), both in the total sample (OR = 0.69, 95%CI = 0.51–0.92, *p* = 0.011) and in women (OR = 0.64, 95%CI = 0.45–0.91, *p* = 0.013), but not in men (OR = 0.74, 95%CI = 0.44–1.26, *p* = 0.26). 

Regarding the intensity of physical activity, we found a protective effect for depression in those individuals who practiced light intensity (crude and after adjusting for sex, age and BMI) and a trend association for those showing moderate intensity. However, no association with vigorous physical activity was found (light intensity: OR = 0.59, 95%CI = 0.39–0.87, *p* = 9.68 × 10 − 3; moderate intensity: OR = 0.73, 95%CI = 0.51–1.04, *p* = 0.083; vigorous intensity: OR = 0.89, 95%CI = 0.44–1.65, *p* = 0.734). We observed similar results in women (light intensity: OR = 0.57, 95%CI = 0.35–0.89, *p* = 0.018; moderate intensity: OR = 0.65, 95%CI = 0.41–1.00, *p* = 0.053; vigorous intensity: OR = 1.23, 95%CI = 0.49–2.66, *p* = 0.630), but not in men (data shown in Table 2).

Finally, when we explored the effect of hours of physical activity on depression in the total sample, we did not find any association neither in crude analyses nor after adjusting for age, sex and BMI (OR = 0.99, 95%CI = 0.97–1.01, *p* = 0.240). These results remained non-significant when women and men were analysed separately (data available in Table 2). 

### 3.4. The BDNF Val66Met Polymorphism and Physical Activity

When exploring the association between Val66Met genotype and physical activity, we found a statistically significant association between practicing physical activity and carrying the Met allele. This association was found in the total sample (OR = 1.26, 95%CI = 1.08–1.47, *p* = 0.002) and in men (OR = 1.48, 95%CI = 1.19–1.84, *p* = 5.28 × 10 − 4), but not in women. The association between carrying the Met allele and the intensity of physical activity was also assessed, but we did not find any significant effect, except for moderate physical activity only in males (OR = 1.41, 95%CI = 1.14–1.76, *p* = 1.74 × 10 − 3). Additionally, we did not find a significant association between hours of physical activity and carrying the Met allele in the total sample, nor in women or men. The associations between the *BDNF* Val66Met polymorphism and physical activity are shown in Appendix A.

### 3.5. Interaction between the BDNF Val66Met Polymorphism, Depression and Hours of Exercise

We found a significant interaction effect between the number of hours of physical activity and the risk for depression conferred by the Val66Met polymorphism (see Table 3 and Figure 1). Thereby, as the number of hours of exercise increases, the Met alleles carriers have a lower risk of depression compared to Val/Val homozygous. Those results remained statistically significant after adjusting for age, sex and BMI (OR = 0.95, 95%CI = 0.90–0.99, *p* = 0.027) and was strengthened in women, also after adjusting for age and BMI (OR = 0.93, 95%CI = 0.87–0.98, *p* = 0.019). However, this interaction effect was not found in men. 

In contrast, when we assessed the binomial physical activity variable or the intensity of physical activity, no significant interactions were found (see Table 3).

## 4. Discussion

The main aim of this study was to determine the potential role of the *BDNF* Val66Met polymorphism in the relationship between depression and physical activity in a large sample of community-based adults.

### 4.1. The BDNF Val66Met Polymorphism and Depression

We did not find a significant association between the *BDNF* Val66Met polymorphism and depression. This result is in line with the ones reported from previous studies, including extensive meta-analyses [31,33]. However, it has been suggested that more accurate assessments of the samples considering different parameters, such as sex, age, ethnicity and gene–gene interactions would be required in order to unveil the peculiarities of this relationship [52].

### 4.2. Depression and Physical Activity

In relation to physical activity, we found an association with a lower prevalence of depression. These results remained significant even after considering classical parameters involved in depression, i.e., sex and age, and also BMI, a variable linked to depression and physical activity [53]. Interestingly, in the PISMA-ep cohort, the prevalence of depression was significantly higher in those participants that reported no practice of physical activity versus those who practiced exercise. This effect was shown in the total sample and in women, but not in men. In this sense, multiple cross-sectional studies have shown significant associations between practicing physical activity and less prevalence of depression [54,55,56]. Moreover, prospective studies have described a significant effect of physical activity preventing the onset of depression. A systematic review [9] and a recent meta-analysis of prospective studies [10] have led to the conclusion that the practice of physical activity is associated with lower odds of incident depression. In their systematic review of prospective studies performed in the general population, Mammen and Faulkner observed in 25 out of 30 studies that reported physical activity at the beginning of the studies was inversely associated with incident depression at follow-up. Similarly, another recent meta-analysis reported an increased risk of developing depression when sedentary behaviour was higher [11]. These results support the effect observed in our cohort, which suggests a potential differential role of physical activity depending on the gender. Interestingly, four of these studies found this inverse association between practice of physical activity and incident depression at follow-up in women, but not in men [15,57,58,59]. Similarly, regarding the intensity of physical activity, we found significant differences in the prevalence of depression between participants who practiced light-intensity physical activity and the sedentary ones, suggesting a protective effect of physical activity, in the total sample and in women. This result is highly interesting, since the implication of light-intensity physical activity in depression has been less studied than moderate- and vigorous-intensity physical activity [60]. Our findings have important implications for public health, since they empathise that practicing light-intensity physical activity seems to have an important effect on depression. Light-intensity physical activity is well-accepted in general and clinical populations and it has properties that make it more transferable for certain population groups such as the elderly. In this sense, further research in this field should be encouraged. 

Regarding high-intensity physical activity, we have previously reported a significant association with a lower prevalence of depression in an extended sample from the PISMA-ep cohort [61]. However, we could not find the same association here, possibly due to the smaller sample used in this study.

### 4.3. The BDNF Val66Met Polymorphism and Physical Activity

When we assessed the association between the Val66Met polymorphism and the practice of physical activity, we found a higher proportion of physically active participants among Met carriers, compared to homozygous Val/Val. Although this polymorphism might be one among many genetic variants influencing a complex behavioural outcome such as physical activity, it is worth highlighting this finding. There are few studies investigating this relationship, the majority reporting no association [62,63,64,65], whereas one study reported that Val/Val individuals experienced higher exertion than Met carriers, arguing that this could influence adherence to exercise [66]. Another study reported similar results regarding differences in intrinsic motivation during exercise in regular exercisers, finding greater intrinsic motivation in Met carriers compared to Val/Val participants [67]. This evidence makes us hypothesise that, if there is an effect, it could be mediated by intermediate factors.

### 4.4. Interaction between the BDNF Val66Met Polymorphism, Depression and Hours of Exercise 

Interestingly, we found an interaction effect between the number of hours of physical activity and a decreased risk of depression, only in Met allele carriers. This interaction suggests that the practice of physical activity exerts a dose-dependent protective effect on the risk of depression, moderated by the *BDNF* Val66Met genotype. The practice of physical activity was significantly associated with less risk of depression in Met allele carriers but not in Val/Val homozygous. This effect was found in the total sample and was strengthened in women. These findings are similar to the results from previous research in healthy adolescent women, which reported an association between physical activity and the level of depressive symptoms, also moderated by the Val66Met genotype [42]. Similarly, they also found that women Met allele carriers who practiced physical activity were associated with lower depressive symptoms, compared to the Val/Val homozygous. This interaction could be explained under the hypothesis of differential susceptibility [68]. According to this, individuals with genetic susceptibility for a certain condition would be more malleable, i.e., would benefit more from a favourable environment. In this sense, risk allele carriers (Met allele), under beneficial conditions (practicing physical activity), would have less risk of depression than those without a certain genetic risk.

The reasons underlying sex differences in this interaction effect remain unclear, although it has been suggested that the social aspects of physical activity (e.g., being encouraged by others, practicing physical activity with family members) have a more beneficial effect on women than in men, and, consequently, this could possibly lead to the observation of less prevalence of depression in physically active women [55,69]. Another potential hypothesis to explain gender differences in the effect of physical activity in depressive symptoms in adults would be the role of oestrogen, which has a key role in physical activity and in mood. In this regard, a recent meta-analysis has reported that physical activity, even in light intensities, is associated with a reduction in depressive symptomatology in adult women of ages around the menopausal transition, repeatedly associated with increased risk of depressive symptoms [70].

In contrast, there is high heterogeneity in studies assessing the potential role of the *BDNF* Val66Met polymorphism on the effect of physical activity on depression. For instance, one cross-sectional study evaluated the role of this polymorphism on the effect of self-reported physical activity on depressive symptoms (using the Center for Epidemiology Depression Scale) in a population-based cohort, not finding a statistically significant moderation effect, possibly due to the sample size [16]. Similar results were found in another study including a sample of 1196 adolescents [71]. Other studies have shown opposite results when comparing between endurance athletes (*n* = 55) and a control group (*n* = 58). The findings showed worse depressive symptoms in the control group participants that were Val/Val homozygous, whereas there were no differences when comparing genotypes in athletes [38]. In a cohort of US veterans, Pitts et al. suggested that the initially observed reduction in cognitive functioning associated with depression would be moderated both by the Val66Met polymorphism and by the practice of physical activity. Thus, in participants with depression, those who practiced physical activity outperformed their not physically active counterparts in different domains of cognition [39]. Interestingly, among Met allele carriers with depression, physically active participants scored better results in subjective cognition, visual learning and work memory tasks in comparison with physically inactive participants.

In experimental studies that analyse interventions with physical activity considering *BDNF* Val66Met genotype, heterogeneous results have also been described. In a 12-week intervention in patients with depression, Rahman et al. observed that the highest proportion of responders to the intervention with physical activity (those who experimented a reduction greater than 50% on the Montgomery Asberg Depression rating scale) were Met allele carriers who were not exposed to childhood adversity, compared to Val/Val homozygous [41]. Furthermore, in a year-lasting intervention with physical activity (aerobic, strength, flexibility and balance training) performed in sedentary community-dwelling participants, Dotson et al. found a decrease in somatic symptoms of depression (one of the four factors of the Center for Epidemiologic Studies Depression Scale) which was more evident in Met allele carriers, but exclusively in men [40]. 

One strength of this study is the extensive and detailed characterization of our community sample. However, due to the cross-sectional design of the study, we are not able to establish causality, thus further longitudinal studies including larger sample sizes would be required. Furthermore, we are also aware that considering the assessment of physical activity compared to the accelerometer and objective measurements of physical activity, questionnaires may be influenced by the participant mood, recall bias and memory inaccuracy, and also social desirability bias [72]. Therefore, future studies should include additional objective measures for the assessment of physical activity.

Finally, further research would be required to assess whether the effect of physical activity on depression risk in longitudinal studies is moderated by the *BDNF* Val66Met genotype. In this way, we could consider using this polymorphism as a biological marker to predict the effect that physical activity would exert on depression risk. Furthermore, functional studies are necessary to investigate how the effect of physical activity on depression risk could be mediated by this polymorphism. In this regard, it has been hypothesised that the increase in BDNF caused by the practice of physical activity could partially be implied from a decrease in the hippocampal atrophy, therefore, protecting against depression [73].

## 5. Conclusions

In conclusion, our findings provide further evidence of a protective effect of physical activity on the risk of depression. We report a gene–environment interaction effect in which Met allele carriers of the *BDNF* Val66Met polymorphism who are more physically active showed a decreased prevalence of depression. Interestingly, this effect is strengthened in women, which has sex differences implications that should be addressed in future studies. Finally, these findings point to physical activity as a potential approach for the prevention of mental disorders in the general population and can also be considered as a non-invasive adjunct treatment for mental diseases.

## Figures and Tables

**Figure 1 ijerph-19-02068-f001:**
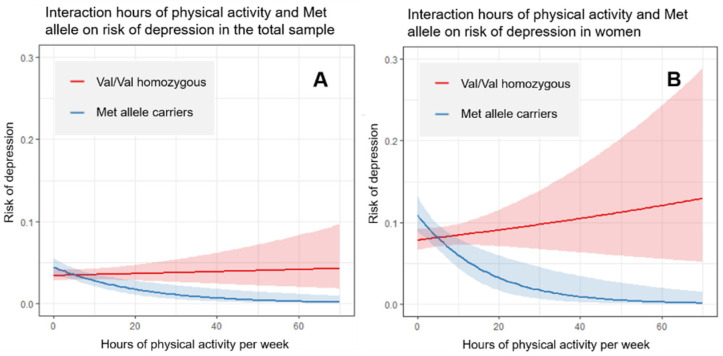
Graphic representation of the interaction between physical activity and the Met allele of the *BDNF* Val66Met polymorphism on the risk of depression, in (**A**) the total sample and (**B**) only in women. Results only in men were not statistically significant.

**Table 1 ijerph-19-02068-t001:** Summary of frequencies of independent variables.

	Total Sample (3123)	Women 1554 (49.76%)	Men 1569 (50.24%)
Mean age (s.d.)	43.18 (15.18)	43.76 (14.94)	42.61 (15.39)
Mean BMI (s.d.)	26.16 (4.50)	25.73 (4.97)	26.59 (3.93)
Diagnosis of depression	No 2914 (93.31%) Yes 209 (6.69%)	No 1406 (90.48%) Yes 148 (9.52%)	No 1508 (96.11%) Yes 61 (3.89%)
Val66Met genotype	ValVal 1947 (62.34%) ValMet 1044 (33.43%) MetMet 132 (4.23%)	ValVal 969 (62.36.%) ValMet 512 (32.95%) MetMet 73 (4.69%)	ValVal 978 (62.33%) ValMet 532 (33.91%) MetMet 59 (3.76%)
Met allele carrying	1176 (37.66%)	585 (37.64%)	591 (37.67%)
Physical activity	No 1260 (40.35%) Yes 1863 (59.65%)	No 676 (43.5%) Yes 878 (56.5%)	No 584 (37.22%) Yes 985 (62.78%)
Mean hours of physical activity (s.d.) *	9.73 (9.79)	9.62 (9.90)	9.83 (9.70)
Intensity of physical activity	No 1260 (40.35%) Light 624 (19.98%) Moderate 988 (31.64%) Vigorous 251 (8.04%)	No 676 (43.5%) Light 350 (22.52%) Moderate 452 (29.09%) Vigorous 76 (4.89%)	No 584 (37.22%) Light 274 (17.46%) Moderate 536 (34.16%) Vigorous 175 (11.15%)

* Mean (s.d.) of reported hours of physical activity was calculated excluding participants reporting no physical activity.

**Table 2 ijerph-19-02068-t002:** Associations between depression and genetic factors or physical activity variables. Statistically significant results are highlighted in bold.

	Total Sample (3123)			Women (1554)			Men (1569)		
	Cases (209)	Controls (2914)	Adjusted * OR (95%CI), *p*	Cases (148)	Controls (1406)	Adjusted ** OR (95%CI), *p*	Cases (61)	Controls (1508)	Adjusted ** OR (95%CI), *p*
Genotypes									
Val/Val	130 (62%)	1817 (62%)		92 (62%)	877 (62%)		38 (62%)	940 (62%)	
Val/Met	66 (32%)	978 (34%)	0.98 (0.71–1.33), 0.891	45 (31%)	467 (33%)	0.96 (0.65–1.40), 0.842	21 (35%)	511 (34%)	1.02 (0.58–1.75), 0.936
Met/Met	13 (6%)	119 (4%)	1.50 (0.78–2.69), 0.195	11 (7%)	62 (5%)	1.71 (0.81–3.32) 0.131	2 (3%)	57 (4%)	0.89 (0.14–3.02), 0.876
Met allele carrying									
Val/Val	130 (62%)	1817 (62%)		92 (62%)	877 (62%)		38 (62%)	940 (62%)	
Met allele carriers	79 (38%)	1097 (38%)	1.04 (0.77–1.39), 0.81	56 (38%)	529 (38%)	1.05 (0.73–1.50) 0.780	23 (38%)	568 (38%)	1.01 (0.59–1.70), 0.974
Physical activity									
No	111 (53%)	1149 (39%)		83 (56%)	593 (42%)		28 (46%)	556 (37%)	
Yes	98 (47%)	1765 (61%)	**0.69 (0.51–0.92), 0.011**	65 (44%)	813 (58%)	**0.64 (0.45–0.91), 0.013**	33 (54%)	952 (63%)	0.74 (0.44–1.26), 0.26
Intensity of physical activity									
No	111 (53%)	1149 (39%)		83 (56%)	593 (42%)		28 (46%)	556 (37%)	
Light	35 (17%)	589 (21%)	**0.59 (0.39–0.87), 9.68 × 10^−3^**	26 (17%)	324 (23%)	**0.57 (0.35–0.89), 0.018**	9 (15%)	265 (18%)	0.64 (0.28–1.32) 0.253
Moderate	52 (25%)	936 (32%)	0.73 (0.51–1.04), 0.083	32 (22%)	420 (30%)	0.65 (0.41–1.00), 0.053	20 (33%)	516 (34%)	0.85 (0.46–1.55), 0.607
Vigorous	11 (5%)	240 (8%)	0.89 (0.44–1.65), 0.734	7 (5%)	69 (5%)	1.23 (0.49–2.66), 0.630	4 (6%)	171 (11%)	0.57 (0.16–1.52), 0.307
Hours of physical activity									
Means expressed in hours	$\bar{x}$ = 4.72, s.d. = 8.16	$\bar{x}$ = 5.88, s.d. = 9.00	0.99 (0.97–1.01), 0.240	$\bar{x}$ = 4.60, s.d. = 8.88	$\bar{x}$ = 5.52, s.d. = 8.84	0.99 (0.97–1.01), 0.378	$\bar{x}$ = 5.00, s.d. = 6.12	$\bar{x}$ = 6.22, s.d. = 9.13	0.98 (0.95–1.01), 0.367

Abbreviations: CI, confidence interval; OR, odds ratio; *p*, *p*-value; $\bar{x}$, mean; s.d., standard deviation. * Adjusted by age, BMI and sex. ** Adjusted by age and BMI. Statistically significant results are highlighted in bold.

**Table 3 ijerph-19-02068-t003:** Interaction between Met allele carriers and physical activity variables on risk of depression. Statistically significant results are highlighted in bold.

		OR (95%CI), *p*		
		Total Sample *	Women **	Men **
Physical activity (yes/no) * carrying Met allele		0.86 (0.48–1.55), 0.616	0.91 (0.44–1.86), 0.786	0.78 (0.27–2.29), 0.641
Intensity of physical activity * carrying Met allele	None	Ref. 1	Ref. 1	Ref. 1
	Light	0.54 (0.22–1.26), 0.167	0.63 (0.22–1.67), 0.363	0.35 (0.05–1.89), 0.257
	Moderate	0.94 (0.46–1.90), 0.858	1.07 (0.43–2.60), 0.881	0.73 (0.21–2.46), 0.607
	Vigorous	2.00 (0.54–8.31), 0.308	1.15 (0.22–6.64), 0.870	4.34 (0.47–96.14), 0.234
Hours of physical activity * carrying Met allele		**0.95 (0.90–0.99), 0.027**	**0.93 (0.87–0.98), 0.019**	0.98 (0.91–1.05), 0.648

Abbreviations: CI, confidence interval; OR, odds ratio; *p*, *p*-value. * Adjusted by age, BMI and sex. ** Adjusted by age and BMI. Statistically significant results are highlighted in bold.

## Data Availability

The data presented in this study are available on request from the corresponding author. Data are not publicly available due to privacy and ethical policies.

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
