# Peer review of "Interaction Effect between Physical Activity and the *BDNF* Val66Met Polymorphism on Depression in Women from the PISMA-ep Study"

_ijerph, 2022, doi:10.3390/ijerph19042068_

Round 1

Reviewer 1 Report

The manuscript by Juan Antonio Zarza Rebollo, et al considers relationship between depression, physical activity and Bdnf val66met Polymorphism. The authors have found the decrease of the risk of depression appearence in women with a cetrain gene-type involved in physical activity from epidemiological stusies in Andalusia, Spain. The result is fresh, importand for practical application and future research. It can help to find an alternative approach to improve heath of population and for mental desease treatment. I would rather reccomend this manuscript for publication in International Journal of Environmental Reasearch and Public  Health, however, after certain changes:

Major comments:

1) Please, modify and commonize the title of the manuscript. The second sentence should be excluded due to it includes very local concepts.

2) Please, provide standard deviations of the means in Supplementary. Also widen the tables in order to show correlation between physical activity and the genotypes. The present form of presentation does not allow to make conclusions presented in the manuscript (decrease of the risk of depression with the increase of PA intensity in Met alleles carriers).  How the data correlates with age and BMI?. 'Met/*' is it mistyping?

3) Please, check dimensions of the values in the Table 3. Does it show Risk of Depression * Carrying Met allele. According to the Table data the risk increases with the PA intensity increase as far as the numbers increase. Please, modify the presentation of the results to make it more understandable.

4) It is not entirely clear why the graphs (Figure 1) show continuous, monotonic curves. Whether tabular data has been approximated?  What mathematical model was used for the approximation? Decipher the abbreviations on the charts and add units of measurement along the axes.

5) Please highlight the importance of the result in Conclusions such as PA has the potential to improve the mental health of the population, and can also be considered as an alternative approach for the treatment of mental deseases.

Reviewer 2 Report

I would like to thank you for submitting and give me the opportunity to review the manuscript entitled: “Physical Activity Moderates the Relationship Between Depresion and the Bdnf val66met Polymorphism in Women. A Gxe Interaction From the Pisma-Ep Study”. The research topic undertaken by the authors is very interesting in times of pandemic due to the lockdown and the reduced physical activity carried out on many occasions, which seems to be related to depressive episodes in the society. These results may be of great importance for increasing the knowledge of the influence and importance of the physical activity, among other variables, in maintaining good mental health in the society. The study is well performed, the results are compelling and adequately presented. Nevertheless, some questions and concerns need to be answered and corrected before the formal acceptance of the manuscript. In this sense, I only have a few minor comments.

In this sense, your results show an association between the Val66Met polymorphism and the practice of physical activity, highlighting in a higher proportion of physically active participants among Met carriers, compared to homozygous Val/Val. In this context, I was wondering whether the lower activity rate found in homozygous Val/Val participants might bias the result of finding only in Met allele carriers and not in homozygous Val/Val, an interaction effect between the number of hours of physical activity and a decreased risk of depression? I would like to know your opinion.

Reviewer 3 Report

This study aimed to look at the moderating effect of the BDNF Val/Met polymorphism on the relationship between physical activity and risk of depression. According the introduction it seems like there is a lack of consensus and the literature is mixed on the role that the BDNF Val/Met polymorphism plays in the relationship between depression and physical activity, thus warranting the need for this study.

Comments:

General

The paper could use some English editing to help make it more readable. There are many little grammar errors and run on sentences. Here is a list of just some of them:

Line 52. Need period after reference 12 and capitalize “These”

Line 55. Should be “others”

Line 75. Should be re-written to read: “..associated with modifying the protein function of BDNF in humans..”

Line 80-81. Should change sentence to read: …”being that the difference in serum BDNF is significantly

Line 85. Sentence structure problem. Try rewording to: “Moreover, the association of this poly-morphism as a mediator of the effect of physical activity on depressive symptoms has been assessed in several studies, also showing conflicting results”

Line 157. Pick one, either use “associations” or “relationships”, but not both in same sentence

Line 270. ”empathise” should be “emphasize”.

Line 301. “Met allele carriers women” should be “women Met allele carriers”

Sentence Lines 179-183 needs to be broken up into 2.

Introduction

The introduction is quite lengthy. I would recommend trimming and making it more streamlined. For instance, there is an abundance of evidence that physical activity is good for depression, thus, a whole paragraph devoted to this, could be eliminated.

Methods

Line 172. What was the rationale for excluding BMI<18.5?

Results

Your interaction is the interaction between hours of PA and Met-allele carriers on the odds of depression. Thus, the title on the top of your figure should be changed to say the interaction of hours of PA and Met-allele on risk of depression in the total sample for Figure 1A and women in Figure 1B. Also, Table 3 is confusing. Again, it is the interaction between PA and Met-allele carrier on risk of depression. Thus, I would have it as “Physical Activity*carrying Met allele”, “Intensity of physical activity*carrying Met allele”, and “Hours of physical activity*carrying Met allele” as your independent variables and “Risk of Depression” as your dependent variable in the Table.

Discussion

Line 241-244. You state how one reason why you may have not have found a relationship between the BDNF VatMet polymorphism was due to not considering different parameters, such as, sex, age, ethnicity and gene-gene interactions. However, in your analysis, you did consider sex and age. Do you have data on the ethnicity of the sample? If so, this should be reported and included as a covariate.

References

A lot of the references are not within the last 5 years. Are there no newer references to report on this topic?

Round 2

Reviewer 1 Report

Almost all my concerns are fully addressed. However, in my opinion, the new title of the article is also not entirely convenient. The authors did not find statistically significant correlations between the studied polymorphism and the risk of depression, so there is no direct relationships that can be moderated. I recommend to modify  the title. Other than that, I have no comments.
